**Subject Category:**
Biology (whole organism)

behaviour

hissing behaviour, nest defence, parental investment, behavioural reaction norms, trade-offs, reproductive cost

**Author for correspondence:**
Bert Thys
e-mail: bert.thys@uantwerpen.be

# Nest defence behavioural reaction norms: testing life-history and parental investment theory predictions

Bert Thys[1], Yorick Lambreghts[1], Rianne Pinxten[1,2] and Marcel Eens[1]

[1]Department of Biology, Behavioural Ecology and Ecophysiology Group, University of Antwerp, Wilrijk, Belgium
[2]Faculty of Social Sciences, Antwerp School of Education, University of Antwerp, Antwerp, Belgium

BT, 0000-0001-7258-1631

Predation is the primary source of reproductive failure in many avian taxa and nest defence behaviour against predators is hence an important aspect of parental investment. Nest defence is a complex trait that might consistently differ among individuals (personality), while simultaneously vary within individuals (plasticity) according to the reproductive value of the offspring. Both complementary aspects of individual variation can influence fitness, but the causality of links with reproductive success remains poorly understood. We repeatedly tested free-living female great tits (*Parus major*) for nest defence (hissing) behaviour across the nesting cycle, by presenting them with a model predator. Hissing behaviour was highly repeatable but, despite population-level plasticity, we found no support for individual differences in plasticity. Path analysis revealed that repeatable differences in hissing behaviour had no direct effect on nest success or fledgling number. However, our best supported path-model showed that more fiercely hissing females laid smaller clutches, with clutch size in turn positively influencing fledgling number, suggesting that females are most likely facing a trade-off between investment in nest defence and reproduction. Strong stabilizing selection for optimal plasticity, in combination with life-history trade-offs, might explain the high repeatability of nest defence and its link with reproductive success.

# 1. Introduction

Predation is the primary source of reproductive failure in many avian taxa and one of the major selection pressures affecting fitness [1,2]. Nest defence against predators is therefore an important aspect of parental investment [3]. Intense nest defence is commonly found to reduce nest predation and increase offspring survival [4–7], while simultaneously entailing costs to the parents in terms of time, energy and risk of injury or death [8,9]. The investment in nest defence is hence expected to be related to how individuals resolve life-history trade-offs, most notably between investment in current versus future reproduction [2,10,11]. From an adaptive perspective, individuals should consistently differ in their risk-taking behaviour towards predators depending on variation in their residual (i.e. future) reproductive value ([12,13], but see also [14]). Individuals with low future fitness expectations (i.e. low residual reproductive value) should favour current reproduction over survival and consistently take more risk in defending their offspring towards predators when compared to individuals with high expectations (i.e. high residual reproductive value) [12]. Specifically, since the investment in current reproduction involves nest defence against predators which is inherently risky, individuals are expected to differ in this investment based on life-history considerations [10,13].

In addition, the intensity of nest defence can be influenced by many other factors, including predation risk, predator type, re-nesting potential and the reproductive value of the current offspring [2,11,15,16]. With regard to the reproductive value of the offspring, parental investment theory predicts that individuals should plastically adjust their investment in nest defence according to the quantity, quality and age of the offspring [3,4,15,17]. Specifically, the optimal level of defence should increase, from newly laid eggs through to fledging, because the relative difference in expected future survival between parents and offspring decreases with offspring age [3,4]. In other words, parents are predicted to invest more in defending nestlings compared to eggs, and older nestlings compared to younger nestlings, because of an increase in the probability that offspring will survive to reproduce [4,15].

Nest defence behaviour is hence a complex trait that may consistently differ among individuals, while simultaneously being adjusted within individuals according to the reproductive value of the offspring. Behavioural reaction norms (BRNs) are ideally suited to describe such variation as these functions relate behavioural phenotypes to environmental variables. Specifically, BRNs describe how behaviour changes over an environmental gradient, where each individual is characterized by a certain combination of intercept (its average level of behaviour) and slope (its behavioural plasticity) [18,19]. Consistent individual differences in average behaviour (i.e. individual variation in BRN intercept) are widespread and commonly referred to as personality [19]. Moreover, accumulating evidence indicates that individuals within the same population can also differ in their behavioural plasticity (i.e. individual variation in BRN slope) [20,21]. Nonetheless, BRNs have rarely been used to describe phenotypic variation in parenting behaviours in general [22,23], or nest defence behaviours in particular [7,24]. For example, a study in tree swallows (*Tachycineta bicolor*) revealed that parents consistently differed in nest defence and in their plastic adjustment of nest defence according to changes in ambient temperature [24]. Interestingly, plasticity in nest defence only appeared to convey reproductive benefits for less aggressively defending males, while reproductive benefits were absent for aggressive males, and females in general [24]. Another study on Ural owls (*Strix uralensis*) showed that consistently more aggressively defending females were more plastic in adjusting their defence according to overwinter changes in food availability and recruited more offspring into the breeding population [7]. These findings highlight that consistent individual differences in nest defence and/or individual specific plasticity in nest defence can influence reproductive success and might have a genetic basis and hence the potential to evolve under selection [18,25,26].

In addition, individual differences in nest defence may translate into fitness in a number of different ways. For example, studies on individual differences in nest defence have found direct effects [7], no effects [27] or sex-specific effects [24] on reproductive success and/or adult survival. However, nest defence might also indirectly influence fitness by interacting with reproductive decisions (e.g. lay date and clutch size), well known to influence reproductive success in birds [28–30]. Hence, understanding the selection mechanisms that may act in maintaining individual differences in nest defence behaviour within populations requires modelling alternative causal scenarios [31–33].

In some cavity-nesting birds, such as tit species (*Paridae*), one form of aggressive nest defence is the so-called hissing behaviour [34]. When a predator approaches the nest cavity, some incubating and brooding females produce loud broadband hissing calls, often accompanied by intense flapping of the wings and lunging at the predator [34]. In the great tit (*Parus major*), hissing behaviour varies greatly among females and has been shown to be repeatable [35–37]. Since hissing behaviour is effective in

startling and deterring predators it is very likely to decrease the chance of nest predation [35,38]. Hissing behaviour can also be costly, since actively lunging at the predator and uttering hissing calls is considered to be risky [35] and, although unexplored to date, likely to entail energetic costs [39,40]. In addition, more passively responding (i.e. non-hissing) females tend to either hide in the nest or (partially) move aside their clutch/brood upon predator confrontation [41], thereby likely reducing female injury and mortality risks, but increasing predation risks for offspring.

Using a population of free-living great tits, we examine individual consistency and individual specific plasticity of female hissing behaviour across a large part of the nesting cycle. Using path analysis, we examine alternative causal pathways linking individual differences in hissing behaviour, reproductive decisions (lay date, clutch size) and reproductive success (fledgling number and mass). Hissing behaviour was quantified by repeatedly presenting breeding females with a taxidermic mount of the great spotted woodpecker (*Dendrocopos major*) inside the nest hole. This species is a natural nest predator of cavity-nesting birds in Eurasia, known to inflict heavy losses on nests (both in the egg and nestling stage), but to impose limited direct mortality risk for adults [42,43]. Based on findings in previous studies [35,37], we predict consistent individual differences in hissing behaviour (i.e. female nest defence personality types). In line with parental investment theory we predict an overall, population level, increase in nest defence intensity across the nesting cycle. Moreover, we expect individual specific plasticity in hissing behaviour across the nesting cycle, with the most tentative hypothesis being that more fiercely hissing females increase their defence more (i.e. being more plastic) according to offspring age compared to less fiercely hissing females [7,22,24]. In addition, we expect that consistently more fiercely hissing females have higher nest success, larger clutches and produce fledglings in better condition, as predicted by life-history and parental investment theory.

# 2. Material and methods

## 2.1. Study population and standard procedures

Data were collected in a population of free-living great tits in the surroundings of Wilrijk, Belgium (51°09′44″ N–4°24′15″ E), during the breeding season of 2016. This population has been monitored since 1997, and at present there are approximately 150 nest-boxes available for great tits to nest in [44–47]. Birds in the population are fitted with a metal leg ring as nestlings or upon first capture, and all adults receive a unique combination of three plastic colour rings, one of which contains a Passive Integrated Transponder tag. Resident birds are aged based on breeding records, and immigrant birds (first-year or older) based on plumage characteristics upon first capture. Some immigrant parents could not be captured due to early stage nest failure and their age remains unknown.

Reproductive activities of all breeding pairs ($N = 107$) were monitored throughout the nesting cycle. Nest-boxes were checked regularly for nest building and subsequently daily to determine lay date. At the end of the laying period and towards the end of incubation, nest-boxes were checked daily to determine the onset of incubation and hatching day, respectively. When 15 days old, nestlings were ringed and weighed to the nearest 0.1 g, used as a proxy for fledgling mass [48]. Finally, nest-boxes were checked after the presumed date of fledging to determine nest success (i.e. whether or not at least one nestling fledged) and fledgling number.

## 2.2. Female hissing behaviour

Hissing trials were performed following procedures described in [37]. In short, the observer quietly approached the nest-box and inserted the head of a taxidermic mount of a great spotted woodpecker into the entrance hole of the focal female's nest-box. The woodpecker was held in this position for 60 s, during which we counted the number of hissing calls produced [35,37]. After the trial, the woodpecker was removed and the lid of the nest-box was opened to confirm the presence and identity of the female.

Hissing tests were performed between April 8 and May 15, between 8.00 and 19.00, on first clutches only. We aimed to perform five repeated trials per breeding female, spanning the entire period female great tits incubate and brood; i.e. three during the incubation stage (days 2, 5 and 9) and two during the early nestling stage when females still brood their nestlings (when nestlings were 2 and 5 days old). A total of 448 tests were performed on 104 breeding females. The number of observations differed between females, either because of premature nest failure or because focal females were (repeatedly) not observed inside the nest-box at the time of testing: five tests ($N$ females = 48), 4 ($N = 45$), 3 ($N = 8$), 2 ($N = 1$), 1 ($N = 2$).

## 2.3. Statistical analyses

### 2.3.1. Sources of variation in hissing behaviour

Random regression [19] was used to model variation in the number of hissing calls in relation to the day in the nesting cycle, the latter expressed as a continuous environmental covariate centered around hatching date (i.e. hatching day = 0). For females that started incubating but abandoned their nest before hatching ($N = 5$), we calculated the expected hatching date (onset of incubation + 12 days). The model included Julian date, time of day and female age as fixed effects and random intercepts and slopes for female ID ($N = 104$). Adjusted repeatability ($R_{adj}$) of hissing behaviour was calculated as the among-individual variance divided by the sum of the among-individual and residual variance [49]. To make results comparable with previous studies [35,37], we also estimated breeding stage-specific repeatabilities from two separate random intercept models using only data from the respective breeding stage. Estimates of fixed effects and intercept–slope covariances were considered significant if their associated 95% credible intervals (CrI) did not overlap zero. However, since variance components are bound to be positive and CrI hence never overlap with zero, we assessed the significance of variances by permutation tests, following methods described in [50]. In short, each permutation hissing data were randomly reshuffled across observations. Random regression analysis as described above was than performed on the new dataset to obtain posterior mean estimates for the variance components of interest. This was repeated 100 times to obtain a 'null' distribution of posterior mean estimates for variances. Next, we calculated the probability (permutation.p) that the observed posterior mean of a variance component was greater than any posterior mean value based on the 'null' distribution [50].

### 2.3.2. Path analysis

Preliminary analysis revealed that nest success was not predicted by reproductive decisions (lay date, clutch size) or repeatable among-individual variation in hissing behaviour (electronic supplementary material, table S1). Hence, for path analyses, we focused on successful nests ($N = 77$) for which we had detailed data on fledgling number and mass. First, a multivariate mixed model was used to partition phenotypic (co)variances of hissing behaviour, reproductive decisions and success into its among-individual (ID) and residual ($R$) matrices [51]. This model was implemented in a Bayesian framework, using Markov Chain Monte Carlo (MCMC) sampling, to ensure that the uncertainty around point estimates in the matrices was appropriately taken forward for subsequent path analysis [31,32,52,53]. The model included hissing behaviour, lay date, clutch size, fledgling number and fledgling mass as response variables (mean-centred, standardized to unit variance and modelled with Gaussian errors) and random intercepts for female ID. Within-individual variance for hissing behaviour was modelled in the residual ($R$) matrix. However, given the absence of repeated measures for reproductive parameters, within-individual (co)variances involving these parameters are not identifiable, and were set to be essentially zero (electronic supplementary material, table S2) [51,53].

Next, path analysis was applied to the estimated ID matrices. Path analysis estimates the partial correlation coefficient between two variables while controlling for effects of all other variables in the model [54]. To obtain posterior mean estimates and CrI for path coefficients, a path analysis was run on each of the estimated ID matrices from the multivariate mixed model (cf. [31,32]). We constructed a set of alternative path-models regarding the causal relationships between hissing behaviour, reproductive decisions and fledgling number and mass. Model fit of these different scenarios was evaluated using AIC values, relative to the model with the lowest AIC value (i.e. $\Delta$AIC) (e.g. [32,33]; but see [55,56]). $\Delta$AIC values greater than two suggest less support for a particular model compared to the model with the lowest AIC value. $\Delta$AIC values greater than two do not falsify an alternative scenario, but simply indicate decreased support compared to the most likely scenario [55,56].

All analyses were performed in R (R Core Team, 2016). The number of hissing calls was mean-centred and standardized to unit variance in all analyses. Mixed models were fitted with the MCMCglmm package [57]. Autocorrelation among samples (less than 0.05 in all cases) and model convergence (scale reduction factor less than 1.05 in all cases; [58]) were carefully assessed. Prior specifications used for analyses are given in the electronic supplementary material, but models were robust to alternative prior specifications [53,57]. Path analyses were performed using the sem package [59].

**Table 1.** Sources of variation in female great tit ($N = 104$) hissing behaviour. Mean estimates for fixed ($\beta$) and random ($\sigma^2$) effects are given with their associated 95% CrI. Fixed effects where CrI do not overlap zero are considered significant and highlighted in italics. Significance of variance components was assessed using permutation tests (see text for details).

|  | hissing calls |
| --- | --- |
| *fixed effects* | $\beta$ (CrI) |
| Intercept | 0.08 (−0.14; 0.33) |
| Day in nesting cycle[a] | *0.02 (0.01; 0.03)* |
| Julian date[b] | −0.05 (−0.22; 0.13) |
| Time of day (mean)[c] | 0.07 (−0.10; 0.25) |
| Time of day (dev)[c] | 0.01 (−0.04; 0.05) |
| Age − adult[d] | −0.14 (−0.50; 0.26) |
| Age − unknown[d] | 0.70 (−0.13; 1.41) |
| *random effects* | $\sigma^2$ (CrI) |
| ID$_{intercept}$ | 0.85 (0.60; 1.12)[#] |
| ID$_{slope}$ | 0.0014 (0.0005; 0.0021)[§] |
| Cov$_{intercepts-slopes}$ | 0.010 (−0.002; 0.021) |
| Residual | 0.19 (0.16; 0.23) |

[a]Centred around hatching day (Hatching day = 0).
[b]Mean test date per individual, in days since July first.
[c]In minutes after sunrise; as the mean test time (mean) and the deviation of each observation from an individuals' mean test time (dev), respectively.
[d]First year as reference category.
[#]$p < 0.001$; [§]$p = 0.12$

# 3. Results

## 3.1. Sources of variation in hissing behaviour

The number of hissing calls produced varied between 0 and 43 (mean ± s.d.; overall dataset: 12.83 ± 11.06; incubation stage: 12.20 ± 11.10; nestling stage: 14.14 ± 10.91). There was strong support for among-individual differences in intercepts (table 1), with hissing behaviour being highly repeatable in the overall dataset ($R_{adj} = 0.81$ [0.75; 0.86]) and within each breeding stage (incubation: $R = 0.79$ [0.72; 0.84]; nestling: $R = 0.76$ [0.65; 0.86]). However, we found no support for the presence of individual differences in slopes, or intercept–slope covariances, indicating the absence of individual differences in plasticity. At the population level, hissing behaviour increased across the nesting cycle (table 1 and figure 1). No support was found for effects of season (mean Julian date of testing per individual), time of day or female age on hissing behaviour (table 1).

## 3.2. Causal links with reproductive fitness

Nest success was not predicted by repeatable among-individual variation in hissing behaviour or lay date and clutch size (electronic supplementary material, table S1). Within successful nests, the single best supported path-model (figure 2; model 3) revealed that nest defence personality type was directly and negatively related to clutch size. Females that started laying later also tended to produce less hissing calls. In addition, lay date had a strong negative effect on clutch size, with the latter positively influencing the number of fledglings. Clutch size also tended to negatively influence average fledgling mass. Direct effects of lay date on fledgling number and mass were not supported (figure 2; model 3). In addition, the second and third best supported path-models, in which hissing behaviour was independent of reproductive decisions (figure 2; model 1) or only related to lay date (figure 2; model 2), found marginally less support.

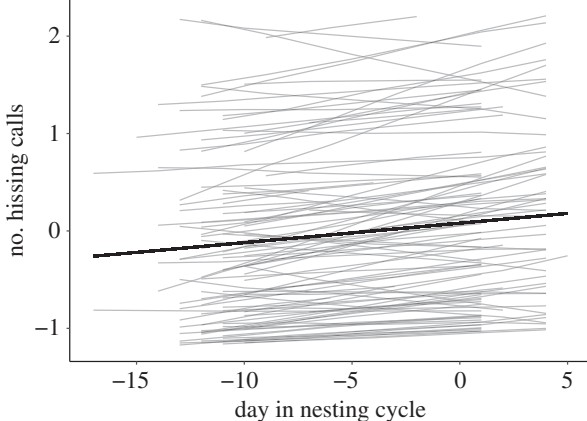

**Figure 1.** Individual (grey lines) and population average reaction norms (black line) of hissing behaviour in female great tits ($N =$ 104) in relation to day in the nesting cycle. The number of hissing calls is mean-centred and standardized to unit variance. Day in the nesting cycle is centred around hatching day (hatching day $= 0$).

## 4. Discussion

We demonstrate that hissing behaviour in free-living female great tits is highly repeatable, providing evidence for the existence of nest defence personality types. Despite population-level plasticity, we found no support for individual differences in plasticity. This indicates that, although females plastically adjust investment in hissing behaviour according to the age of their offspring, there are no differences among females in the way they do so. Interestingly, nest defence personality type did not predict nest success, nor was it directly linked with fledgling number or mass. However, our best supported path-model suggests that consistently more fiercely hissing females laid smaller clutches, with clutch size positively influencing the number of fledglings being produced. Together, our findings suggest that nest defence personality type does not predict overall nest success, but that females are faced with a trade-off between the investment in nest defence and clutch size, and hence number of fledglings produced.

### 4.1. Sources of variation in hissing behaviour

Our finding that female great tits show consistent differences in hissing behaviour extends previous results in great tits during the incubation stage [32–34] and adds to growing evidence for the existence of nest defence personality types in a variety of bird species [7,24,27,41,60]. Since repeatability was estimated using data from a single breeding season, the observed among-individual differences in hissing behaviour may be (partially) caused by local environmental effects (e.g. differences in territory characteristics) in addition to individual characteristic of the female [61,62]. Disentangling the relative contribution of these effects requires data across multiple breeding seasons, which we are currently lacking. Nest defence behaviour is however commonly found to be repeatable across years [7,24,27,41,60]. Moreover, since repeatability is generally thought to set an upper limit to heritability, hissing behaviour might have underlying additive genetic variation and hence the potential to evolve under selection [26], which however remains to be determined.

In line with parental investment theory, we found support for population-level plasticity in hissing behaviour, with an overall increase in investment in hissing behaviour across the nesting cycle. Such an increase with offspring age has been found in a variety of bird species [5,6,27,63] and suggests that birds are able to adjust their nest defence intensity according to the reproductive value of their offspring [9,15]. However, there are a number of alternative explanations for the observed population-level increase in nest defence. First, an increase in nest defence across the nesting cycle has also been explained by positive reinforcement and loss of fear with repeated predator exposure [64]. We cannot exclude this possibility as our standardized sampling design does not allow disentangling the effects of offspring age and repeated predator exposure. Second, the observed population-level increase in nest defence intensity may have resulted from seasonal changes in predation risk and/or re-nesting potential [3,9,15,16]. We consider this unlikely since we found no evidence for seasonal effects on hissing intensity in our overall dataset. However, we did find that, within successful nests, females

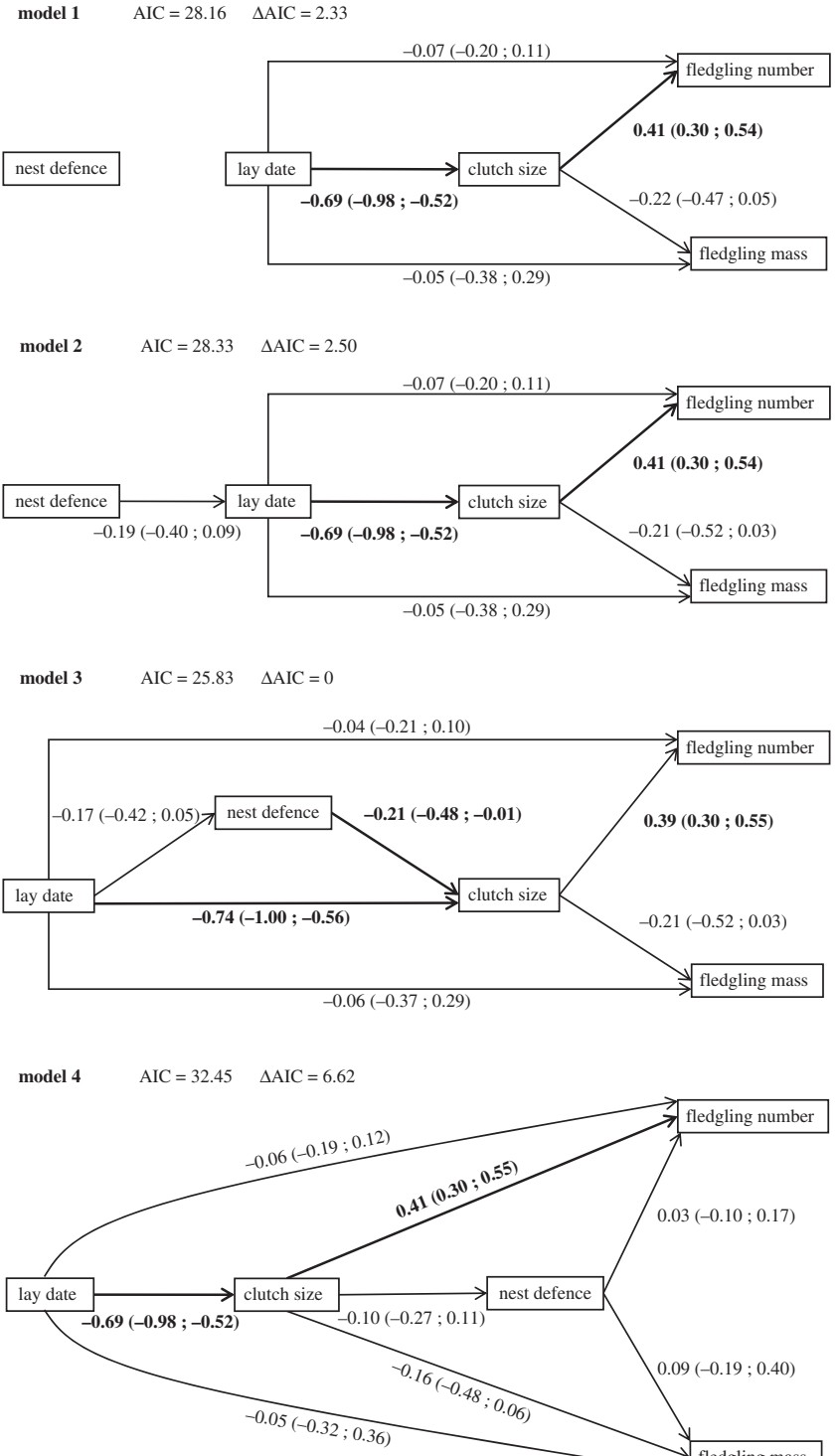

**Figure 2.** Path-models investigating the causal relationships between nest defence personality type (Nest defence), reproductive decisions (lay date, clutch size), fledgling number and average fledgling mass in female great tits ($N = 77$). One-headed arrows depict hypothesized causal relationships, and supported path coefficients (95% CrI) are depicted in bold. Models are presented with their associated ($\Delta$)AIC, and the model with the lowest AIC value was best supported (model 3). A null model (model 0) in which all traits were independent (not depicted) found less support (AIC = 88.24).

that started egg-laying later tended to produce fewer hissing calls, possibly due to reduced predation risk later in the season (e.g. [36]). Finally, the risk of predation by woodpeckers might be higher for nestlings compared to eggs [9,43], potentially resulting in higher female defence levels during the nestling stage compared to the incubation stage. Although hissing intensity also increased within the incubation

stage, arguing against the latter possibility, we cannot entirely exclude the possible role of predation risk in explaining the observed population-level increase.

Despite population-level plasticity in hissing behaviour, we found no support for differences in plasticity among females, indicating that all females responded in a similar way to changes in the reproductive value of their offspring with age. We are confident that this finding is not due to a lack of statistical power, as our sample size corresponds to recommended overall sample sizes for random regression models [65]. Ultimately, the presence of population-level plasticity, but the absence of individual differences in plasticity, might have resulted from strong (stabilizing) selection for an optimal level of plasticity in nest defence according to the age of the offspring, thereby eroding differences in plasticity among individuals. Assuming that plasticity has an underlying genetic basis [18,50], this suggests that adjusting nest defence in response to offspring age might be adaptive, but that the fitness costs and benefits associated with plasticity in nest defence are similar for all females (and by extension among personality types) ([25]; but see [24]). Although we found no evidence for individual differences in plasticity in nest defence across the nesting cycle, such differences may be present along other environmental gradients, potentially influencing fitness [7,24]. Hence, applying BRNs to parenting behaviour across biologically relevant environmental gradients is necessary to adequately describe phenotypic variation and understand how selection might be acting on different levels of variation [22,23].

## 4.2. Causal links with reproductive success?

Both life-history and parental investment theory predict that individual differences in nest defence are linked to reproductive success [12,15,17]. Interestingly, we found that repeatable among-individual differences in hissing behaviour did not predict nest success, suggesting a limited role of hissing behaviour in defending the nest against predation. However, between-year variation in environmental conditions, such as food and predation pressure, likely influence the reproductive consequences of nest defence strategies [2,11,14,16]. In addition, the absence of a relationship between nest success and hissing personality type may be related to the fact that our study was performed in a nest-box population, which provides birds with uniformly dimensioned artificial nesting cavities that may be less prone to predation compared to natural cavities [66,67]. Besides predation, nest success is also typically influenced by other factors, including the quality of both parents in terms of good genes and nestling provisioning [31,68].

Our single best supported path-model revealed that individual differences in hissing behaviour were directly linked with reproductive decisions. It should be noted that our second and third best supported path-models, in which hissing behaviour was independent of reproductive decisions or only related to lay date, respectively, found only marginally less support. Despite finding decreased support, these alternative scenarios cannot be dismissed [56] and further research is hence necessary to validate our results. With this in mind, our best supported path-model does suggest that consistently more fiercely hissing females laid smaller clutches, with clutch size in turn positively influencing the number of fledglings produced. Hence, these results suggest that consistent individual differences in hissing behaviour during early stages in the nesting cycle (i.e. incubation and early nestling stage) likely reflect early investment in reproduction (i.e. clutch size). However, the direction of the relationship suggests that females are faced with a trade-off between investment in hissing behaviour and clutch size. In other words, consistently more aggressively defending females seemingly suffer a reproductive cost in terms of the number of eggs, and hence fledglings, produced. This is in line with findings in two recent studies on songbirds that found evidence for trade-offs between female aggressive behaviours and, at least some, aspects of reproductive investment ([69,70]; see also [71]). These trade-offs are often found to be mediated by individual differences in hormone levels, such as corticosterone and testosterone ([72,73]; but see also [74]). Hence, more aggressively defending female great tits might have higher (stress-induced) corticosterone and testosterone levels, resulting in reduced reproductive investment (but see [75]).

Interestingly, there is some evidence that females that produce hissing calls, compared to females that do not, are less likely to be killed inside their nest cavity [35]. Hence, although more aggressively defending females might face a trade-off with the investment in current reproduction (i.e. smaller clutch size), this reproductive cost might be outweighed (or balanced) by higher survival probabilities. However, as mentioned above, fluctuating environmental conditions may influence reproductive consequences of nest defence strategies, which awaits formal testing for hissing behaviour, using long-term data.

# 5. Conclusion

By applying a behavioural reaction norm approach, we demonstrate very high repeatability of hissing behaviour, as well as population-level plasticity across the nesting cycle. However, individual differences in plasticity were absent, suggesting that the fitness cost and benefits associated with plasticity in hissing behaviour across the nesting cycle are similar for all females. In addition, following our best supported path-model, nest defence personality type most likely relates directly to the investment in current reproduction, with consistently more fiercely defending females laying smaller clutches, suggesting a trade-off between investment in aggressive nest defence and clutch size. Overall, strong stabilizing selection for an optimal level of plasticity, together with life-history trade-offs, might result in the high repeatability of hissing behaviour and its link with reproductive success. Long-term data are necessary to assess (1) whether reproductive consequences of nest defence personality types differ among breeding attempts, potentially due to fluctuations in food availability and predation pressure and (2) whether individuals that are better able to plastically adjust their nest defence strategies to environmental conditions have higher reproductive success. Furthermore, experimental studies that manipulate reproductive investment (e.g. clutch and brood size manipulations), while simultaneously applying a BRN approach, will vastly improve our understanding of the adaptive evolution of both consistency and individual specific plasticity in nest defence behaviour, and parenting behaviour, in general.

Ethics. Experiments were approved by the ethical committee of the University of Antwerp (ID 2014-88 and ID 2017-23), performed in accordance with Belgian and Flemish laws, and adhere to ASAB/ABS guidelines. The Royal Belgian Institute of Natural Sciences provided ringing licences for all authors and technicians.

Data accessibility. The dataset supporting this article has been uploaded as part of the supplementary material (electronic supplementary material, table S3).

Authors' contributions. B.T., R.P. and M.E. conceived and designed the study. B.T. and Y.L. performed the fieldwork and processed the data. B.T. performed the statistical analyses and took the lead in writing the first draft, with important input from Y.L. All authors contributed in revising the manuscript and gave final approval for publication.

Competing interests. We have no competing interests.

Funding. This study was made possible through financial support from the University of Antwerp (to B.T., Y.L., R.P. and M.E.) and the FWO Flanders through a PhD fellowship to B.T. (grant ID: 1.1.434.18N) and a FWO-project to R.P. and M.E. (project ID: G0A36.15).

Acknowledgements. We thank Gilles de Meester, Geert Eens, Thomas Raap and Peter Scheys for important support during fieldwork, and Stefan Van Dongen and Yimen Araya-Ajoy for statistical advise. Special thanks to Anne Charmantier, Andrea Grunst, Melissa Grunst, Arne Iserbyt, Thomas Raap and two anonymous reviewers for their comments on an earlier version of the manuscript. The Royal Belgian Institute of Natural Sciences kindly provided stuffed specimens of the great spotted woodpecker.

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
