## [Reviewer comments · Royal Society Open Science]

Review History

RSOS-182180.R0 (Original submission)

Review form: Reviewer 1

Is the manuscript scientifically sound in its present form?

Yes

Are the interpretations and conclusions justified by the results?

No

Is the language acceptable?

No

Is it clear how to access all supporting data?

Yes

Do you have any ethical concerns with this paper?

No

Have you any concerns about statistical analyses in this paper?

No

Recommendation?

Major revision is needed (please make suggestions in comments)

Comments to the Author(s)

I reviewed this manuscript previously when it was submitted to Proceedings of the Royal Society B. Generally, I think the majority of the concerns raised in the initial review were adequately addressed with the revisions, with one important exception; the interpretation of the strength of support for the alternative path analysis models. Models 1-3 are effectively all equally well supported (within 3 AIC of best model), and they each provide very different interpretations of the effect of hissing behaviour on breeding parameters and nest success. I think the only conclusion that can be drawn from this is that the role of hissing behaviour in shaping these other parameters is totally unclear. Although the authors vaguely acknowledge this in the discussion, they go on to primarily interpret the results of model 3. Further, the abstract presents the results of model 1 without clarifying that there are two alternative interpretations that are equally well supported, including no relationship between hissing and any of the other parameters.

Review form: Reviewer 2

Is the manuscript scientifically sound in its present form?

Yes

Are the interpretations and conclusions justified by the results?

Yes

Is the language acceptable?

Yes

Is it clear how to access all supporting data?

Yes

Do you have any ethical concerns with this paper?

No

Have you any concerns about statistical analyses in this paper?

No

Recommendation?

Accept as is

Comments to the Author(s)

I am very happy with the changes - well done.

I have no further concerns.

Decision letter (RSOS-182180.R0)

20-Feb-2019

Dear Mr Thys,

The editors assigned to your paper ("Nest defence behavioural reaction norms: testing life history and parental investment theory predictions") have now received comments from reviewers. We would like you to revise your paper in accordance with the referee and Associate Editor suggestions which can be found below (not including confidential reports to the Editor). Please note this decision does not guarantee eventual acceptance.

Please submit a copy of your revised paper before 15-Mar-2019. Please note that the revision deadline will expire at 00.00am on this date. If we do not hear from you within this time then it will be assumed that the paper has been withdrawn. In exceptional circumstances, extensions may be possible if agreed with the Editorial Office in advance. We do not allow multiple rounds of revision so we urge you to make every effort to fully address all of the comments at this stage. If deemed necessary by the Editors, your manuscript will be sent back to one or more of the original reviewers for assessment. If the original reviewers are not available, we may invite new reviewers.

- Data accessibility

If you wish to submit your supporting data or code to Dryad (<http://datadryad.org/>), or modify your current submission to dryad, please use the following link:
<http://datadryad.org/submit?journalID=RSOS&manu=RSOS-182180>

- **Competing interests**

- **Authors' contributions**

- **Acknowledgements**

- **Funding statement**

on behalf of Dr Alexander Ophir (Associate Editor) and Kevin Padian (Subject Editor)
openscience@royalsociety.org

Associate Editor's comments (Dr Alexander Ophir):

Dear Dr. Thys,

I have received the comments from two expert reviewers and as you can find they were both pleased by the degree to which you addressed issues raised in the past. However one reviewer continues to emphasize the importance of more equitably discussing the three possible models

that received relatively equal support. The reviewer acknowledges that you have now mentioned this point, but they indicate that you have not done this sufficiently, such that you appear to favor one interpretation over others that statistically are equally good. I think you should more directly address this issue. There are several way sin which I think you could accomplish this, for example, you could walk back your interpretation and/or provide deeper discussion of the other ways to interpret the results based on the other models, or you can provide justification for why your preferred model deserves more prominence. Whatever you decide, I think that it is an important enough issue that it merits some more thought. This point notwithstanding, you have produced a nice study that clearly has generated some positive interest and I look forward to receiving your responses.

Editor comments:

It appears that the reviewers are generally satisfied with the revisions, with one exception detailed by the AE. Please address this in your next version, and thanks for submitting.

Comments to Author:

Reviewers' Comments to Author:

Reviewer: 1

Comments to the Author(s)

I reviewed this manuscript previously when it was submitted to Proceedings of the Royal Society B. Generally, I think the majority of the concerns raised in the initial review were adequately addressed with the revisions, with one important exception; the interpretation of the strength of support for the alternative path analysis models. Models 1-3 are effectively all equally well supported (within 3 AIC of best model), and they each provide very different interpretations of the effect of hissing behaviour on breeding parameters and nest success. I think the only conclusion that can be drawn from this is that the role of hissing behaviour in shaping these other parameters is totally unclear. Although the authors vaguely acknowledge this in the discussion, they go on to primarily interpret the results of model 3. Further, the abstract presents the results of model 1 without clarifying that there are two alternative interpretations that are equally well supported, including no relationship between hissing and any of the other parameters.

Reviewer: 2

Comments to the Author(s)

I am very happy with the changes - well done.
I have no further concerns.

Author's Response to Decision Letter for (RSOS-182180.R0)

See Appendix A.

RSOS-182180.R1 (Revision)

Review form: Reviewer 1

Is the manuscript scientifically sound in its present form?

Yes

Are the interpretations and conclusions justified by the results?

Yes

Is the language acceptable?

Yes

Is it clear how to access all supporting data?

Yes

Do you have any ethical concerns with this paper?

No

Have you any concerns about statistical analyses in this paper?

No

Recommendation?

Accept as is

Comments to the Author(s)

The authors are now explicit about their criteria ($\Delta AIC > 2$) for considering a significant difference between models, but clarify that two models were only marginally worse than this. These revisions to the MS have addressed my earlier comment.

Decision letter (RSOS-182180.R1)

12-Mar-2019

Dear Mr Thys,

I am pleased to inform you that your manuscript entitled "Nest defence behavioural reaction norms: testing life history and parental investment theory predictions" is now accepted for publication in Royal Society Open Science.

Please provide as soon as possible in a zip file:

- 1) individual figure files for each figure
- 2) individual table files for each table
- 3) a caption file containing captions for the tables and figures
- 4) an editable version of the manuscript (Word or Latex preferred).

Once we have the above, you can expect to receive a proof of your article in the near future. Please contact the editorial office (openscience_proofs@royalsociety.org) and

openscience@royalsociety.org) to let us know if you are likely to be away from e-mail contact. Due to rapid publication and an extremely tight schedule, if comments are not received, your paper may experience a delay in publication.

on behalf of Dr Alexander Ophir (Associate Editor) and Kevin Padian (Subject Editor)
openscience@royalsociety.org

Reviewer comments to Author:
Reviewer: 1

Comments to the Author(s)
The authors are now explicit about their criteria ($\Delta AIC > 2$) for considering a significant difference between models, but clarify that two models were only marginally worse than this. These revisions to the MS have addressed my earlier comment.

Appendix A

Detailed response to the editor and referees

Reference number: RSOS-182180

Manuscript title: Nest defence behavioural reaction norms: testing life history and parental investment theory predictions

Authors: Bert Thys, Yorick Lambregts, Rianne Pinxten, and Marcel Eens

Associate Editor's comments (Dr Alexander Ophir):

Dear Dr. Thys,

I have received the comments from two expert reviewers and as you can find they were both pleased by the degree to which you addressed issues raised in the past. However one reviewer continues to emphasize the importance of more equitably discussing the three possible models that received relatively equal support. The reviewer acknowledges that you have now mentioned this point, but they indicate that you have not done this sufficiently, such that you appear to favor one interpretation over others that statistically are equally good. I think you should more directly address this issue. There are several ways in which I think you could accomplish this, for example, you could walk back your interpretation and/or provide deeper discussion of the other ways to interpret the results based on the other models, or you can provide justification for why your preferred model deserves more prominence. Whatever you decide, I think that it is an important enough issue that it merits some more thought. This point notwithstanding, you have produced a nice study that clearly has generated some positive interest and I look forward to receiving your responses.

Editor comments:

It appears that the reviewers are generally satisfied with the revisions, with one exception detailed by the AE. Please address this in your next version, and thanks for submitting.

Response:

We like to kindly thank the Editor and Associate Editor for the time and effort in assessing our manuscript. Moreover, we thank the Associate Editor for his suggestions on how to directly address the remaining issue raised by Reviewer 1. Below we outline how we have dealt with this outstanding comment.

Kind regards,

Bert Thys (on behalf of all authors)

Comments to Author:

Reviewers' Comments to Author:

Reviewer: 1

Comments to the Author(s)

I reviewed this manuscript previously when it was submitted to Proceedings of the Royal Society B. Generally, I think the majority of the concerns raised in the initial review were adequately addressed with the revisions, with one important exception; the interpretation of the strength of support for the alternative path analysis models. Models 1-3 are effectively all equally well supported (within 3 AIC of best model), and they each provide very different interpretations of the effect of hissing behaviour on breeding parameters and nest success. I think the only conclusion that can be drawn from this is that the role of hissing behaviour in shaping these other parameters is totally unclear. Although the authors vaguely acknowledge this in the discussion, they go on to primarily interpret the results of model 3. Further, the abstract presents the results of model 1 without clarifying that there are two alternative interpretations that are equally well supported, including no relationship between hissing and any of the other parameters.

Response:

First of all, we like to thank the reviewer for the time and effort in re-assessing our manuscript, as well as the valuable comments during the initial review which vastly improved our manuscript. We are also pleased to hear that the reviewer thinks we adequately addressed the concerns raised in the initial review, notwithstanding the exception outlined above.

Regarding the comment outlined above, we acknowledge that a solid description of the interpretation of (Δ)AIC was lacking in the previous version of the manuscript. In general (see e.g. [1-3]), models that have an AIC value within two of the model with the lowest AIC value are considered to be equally supported and can hence be considered equivalent. However, models with Δ AIC values greater than two relative to the model with the lowest AIC value suggest decreased support. Of course this does not mean that these models do not find any support, or the falsification of these models, but simply indicates that they find less (decreased) support, relative

to the model with the lowest AIC value. We now explicitly clarify this in the statistics section (L186-191). Moreover, throughout the manuscript (including the abstract, L22-23) we now explicitly mention that it is our best supported path-model (i.e. model 3) that suggests that hissing behavior most likely relates to clutch size (see also L248-250; L343-346). We believe this to be not overstating the results, given we can effectively differentiate between model 3 and models 1-2 based on ΔAIC (>2), and model 3 therefore does present to best supported (hence most likely) scenario. However, given models 1-2 are within 3 AIC of the best supported model, there is indeed some caution needed for the interpretation of our results. We do agree with the reviewer that in our previous version this was done too vaguely. Therefore, we now more explicitly mention in our discussion that the model in which hissing behaviour is independent of reproductive decisions (i.e. model 1) or only related to lay date (i.e. model 2) cannot be dismissed, and that further research is therefore necessary to validate our results (please see L312-321).

Reviewer: 2

Comments to the Author(s)

I am very happy with the changes - well done.

I have no further concerns.

Response:

We are very pleased to hear this and like to sincerely thank the reviewer again for the previous constructive comments and feedback. This vastly improved our manuscript.

References used in this response letter

- [1] Burnham KP, Anderson DR. 2002 Model Selection and Multimodel Inferences: a Practical Information-Theoretic Approach. New York, NY: Springer.
- [2] Burnham KP, Anderson DR, Huyvaert KP. 2011 AIC model selection and multimodel inference in behavioral ecology: some background, observations, and comparisons. *Behav Ecol Sociobiol.* 65:23-35.
- [3] Symonds MRE, Moussalli A. 2011 A brief guide to model selection, multimodel inference and model averaging in behavioural ecology using Akaike's information criterion. *Behav Ecol Sociobiol.* 65:13-21.